# Leveraging Contrastive Learning and Knowledge Distillation for Incomplete Modality Rumor Detection

**Fan Xu[1], Pinyun Fu[1], Qi Huang[1], Bowei Zou[2]\*, AiTi Aw[2], Mingwen Wang[1]**

[1] Jiangxi Normal University
[2] Institute for Infocomm Research, A*STAR
{xufan, huangqi, mwwang}@jxnu.edu.cn
fupinyun@163.com
{zou_bowei, aaiti}@i2r.a-star.edu.sg

## Abstract

Rumors spread rapidly through online social microblogs at a relatively low cost, causing substantial economic losses and negative consequences in our daily lives. Existing rumor detection models often neglect the underlying semantic coherence between text and image components in multimodal posts, as well as the challenges posed by incomplete modalities in single modal posts, such as missing text or images. This paper presents CLKD-IMRD, a novel framework for Incomplete Modality Rumor Detection. CLKD-IMRD employs Contrastive Learning and Knowledge Distillation to capture the semantic consistency between text and image pairs, while also enhancing model generalization to incomplete modalities within individual posts. Extensive experimental results demonstrate that our CLKD-IMRD outperforms state-of-the-art methods on two English and two Chinese benchmark datasets for rumor detection in social media[1].

## 1 Introduction

Social media platforms like Twitter and Weibo allow people to contribute vast amounts of content to the Internet. However, this content is often rife with rumors, which can lead to significant societal problems. For instance, in the first three months of 2020, nearly 6,000 people were hospitalized due to coronavirus misinformation[2], while COVID-19 vaccine misinformation and disinformation are estimated to cost 50 to 300 million each day[3].

Humans are generally susceptible to false information or rumors and may inadvertently spread them (Vosoughi et al., 2018). Suffering from the low coverage and long delay of rumor detection manually, automatic rumor detection models are essential. Previous text modality-based rumor detection models focused on exploring propagation (Lao et al., 2021), user information (Li et al., 2019), and writing styles (Przybyla, 2020; Xu et al., 2020).

Furthermore, as reported in Jin et al. (2017), microblogs with pictures have been found to have 11 times more access than those without pictures, which highlights the importance of multimodal content in rumor detection. Specifically, some studies have investigated the fusion of different modalities, such as images and text, by directly concatenating their representations (Khattar et al., 2019; Singhal et al., 2022). Nevertheless, directly concatenating two different modalities (i.e., textual and visual modalities) may not capture their deep semantic interactions. To address this issue, co-attention-driven rumor detection models aim to extract the alignment between the two modalities (Wu et al., 2021; Zheng et al., 2022). However, current multimodal rumor detection models typically overlook the problem of incomplete modalities, such as the lack of images or text in a given post, making them less effective in handling such cases.

To address the above issues, we propose a novel approach, Contrastive Learning and Knowledge Distillation for Incomplete Modality Rumor Detection (CLKD-IMRD). In fact, supervised contrast learning effectively pulls together representations of the same class while excluding representations from different classes, and knowledge distillation can effectively handle the incomplete modality cases when debunking rumors. More specifically, we first construct a teacher model that consists of multimodal feature extraction, multimodal feature fusion, and contrastive learning. Then, we adopt knowledge distillation to construct

---

\* Corresponding author.

[1] Code for all experiments in this paper are available at https://github.com/fupinyun/CLKD-IMRD

[2] https://www.who.int/news-room/feature-stories/detail/fighting-misinformation-in-the-time-of-covid-19-one-click-at-a-time

[3] https://asprtracie.hhs.gov/technical-resources/resource/11632/covid-19-vaccine-misinformation-and-disinformation-costs-an-estimated-50-to-300-million-each-day

a student model that can handle incomplete modalities (i.e., lack of images or texts). Experimental results and visualizations demonstrate that our CLKD-IMRD outperforms state-of-the-art methods on both English and Chinese datasets in rumor detection on social media.

This paper makes three major contributions.

(1) We propose a novel rumor detection framework that integrates the supervised contrastive learning into our teacher network. This framework captures the deep semantic interactions among source texts, images, and user comments simultaneously.

(2) We present a knowledge distillation driven rumor detection model that can handle incomplete modalities (i.e., lack of images or texts), which is a common phenomenon on social media.

(3) We conduct extensive experiments and visualization to verify the effectiveness of the proposed CLKD-IMRD model on the four benchmark rumor corpora compared to many strong baseline models.

## 2 Related Work

### 2.1 Single Modality-based Models

Generally, propagation patterns are good indicators of rumor detection, because the interaction among different kinds of users (e.g., normal users and opinion leaders) or a source microblog and its subsequent reactions can help to detect rumor. As mentioned in (Wu et al., 2015), a rumor was first posted by normal users, and then some opinion leaders supported it. Finally it was reposted by a large number of normal users. The propagation of a normal message, however, was very different from the propagation pattern of a fake news. The normal message was posted by opinion leaders and was reposted directly by many normal users instead. Ma et al. (2017) also proposed a propagation path-based model to debunk rumors. Lao et al. (2021) presented a propagation-based rumor detection model to extract the linear temporal sequence and the non-linear diffusion structure simultaneously.

In addition, user information is also a good insight to rumor detection, because the authoritative users are unlikely to publish a rumor, while normal users have a high probability to produce or repost a rumor instead. Specifically, Mukherjee and Weikum (2015) assessed user credibility based on several factors, including community engagement metrics (e.g., number of answers, ratings given, comments, ratings received, disagreement and number of raters), inter-user agreement, typical perspective and expertise, and interactions. Yuan et al. (2020) proposed a fake news detection model based on integrating the reputation of the publishers and reposted users. Li et al. (2019) presented a user credit driven multi-task learning-based framework to conduct rumor detection and stance detection simultaneously.

Furthermore, the writing style is a crucial factor in rumor detection, as there are often distinct differences in the vocabulary and syntax used in rumors compared to non-rumors. Specifically, Rubin et al. (2015) proposed a rhetorical structure-based framework for news verbification. The rhetorical structure is a wildly used theory in discourse analysis, which describes how the constituent units of a discourse are organized into a coherent and complete discourse according to a certain relationship. According to the rhetorical structure theory, the relationship between discourse units mostly shows the nucleus-satellite relationship. Compared with the author's communicative intention, the discourse unit in the core position is in a relatively important position, and there are different discourse relationships between the core discourse unit and the satellite discourse unit. Potthast et al. (2018) adopted some lexical features (e.g., characters uni-gram, bi-gram and tri-gram, stop words, part of speech, readability value, word frequency, proportion of quoted words and external links, number of paragraphs, and average length of a text) for fake news detection. Przybyla (2020) presented a writing style-based fake news detection framework to verify the effectiveness of sentiment vocabularies.

### 2.2 Multimodal-based Models

Wang et al. (2018) presented a GAN-based multimodal fake news detection model to adopt VGG (Visual Geometry Group) to extract visual features and employ a CNN (Convolutional neural network) to extract text features simultaneously. They also integrated the characteristics of the invariance of an event to facilitate the detection of the newly arrived events of fake news. Khattar et al. (2019) proposed a variational auto-encoder-based fake news detection model to capture the shared representation between textual and visual modalities. Their novel decoder can utilize

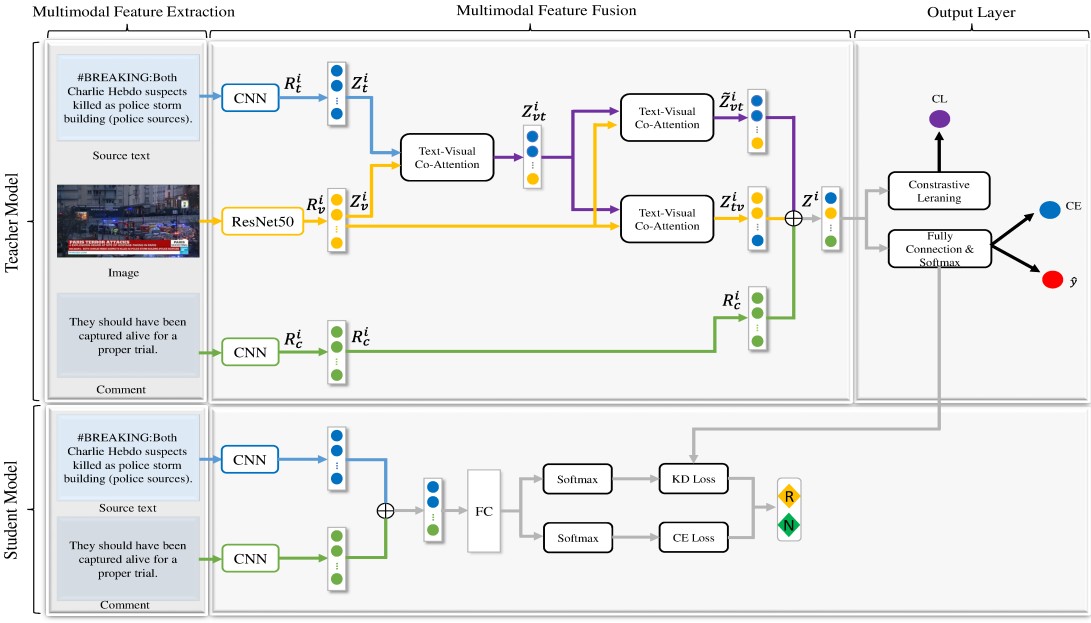

Figure 1: Framework of CLKD-IMRD. FC: fully connection; R: rumor; N: non-rumor.

the multimodal representations from a variational auto-encoder. Zhou et al. (2020) presented a similarity-based fake new detection model to calculate the similarity between multimodal and cross-modal features jointly. In addition, Dhawan et al. (2022) proposed a graph attention-based fake news detection framework to incorporate the interaction between textual word information and the local visual object from images. Wei et al. (2021) presented a graph convolutional networks-based rumor detection model to investigate the reliability of potential relationships in propagation structures through a Bayesian model. Yuan et al. (2019) presented a graph-based rumor detection model to combine the encoding of local semantic and global structural information simultaneously. Zheng et al. (2022) proposed a GAT-based rumor detection model to integrate textual, visual, and social graphs altogether.

Despite the emergence of many multimodal rumor detection models in recent years, they often overlook the distinguishing features of samples with the same or different types of labels. Additionally, these models do not account for incomplete modality situations, where the image may fail to load, which is a common occurrence on the Internet. To address these gaps, we propose a novel framework based on contrastive learning and knowledge distillation to effectively debunk

rumors with incomplete modalities.

## 3 Methodology

### 3.1 Task Formulation

Let's define $P = \{p_1, p_2, ..., p_n\}$ as a set of posts. Each post $p_i$ consists of $\{t_i, v_i, c_i\}$, where $t_i$ indicates a source text, $v_i$ donates an image, and $c_i$ refers to a comment. We approach rumor detection as a binary classification task, with a goal of learning a function $f(p_i) \rightarrow y$, where $p_i$ represents the given multi-modal post, and $y$ represents the label assigned to the post, where $y$=1 indicates a rumor and $y$=0 indicates a for non-rumor.

### 3.2 Framework of CLKD-IMRD

Figure 1 illustrates our proposed CLKD-IMRD framework for rumor detection. Specifically, we employ a multimodal feature extraction module to obtain representations of the source text, images, and comments from a given post. For the teacher model, the extracted multimodal features are fed into a multimodal feature fusion module. During the multimodal feature fusion phase, we adopt visual features to enhance textual features, using the cross-modal joint attention mechanism to obtain enhanced features between textual and visual representations. Then, in the output layer module, we integrate features from different modalities into a supervised contrastive learning framework. For

the student model which lacks visual modality, we directly concatenate the representation of source text and comments, and adopt knowledge distillation to obtain corresponding classification results.

### 3.2.1 Teacher Model

The teacher model is composed of three modules: multimodal feature extraction, multimodal feature fusion, and output layer.

**Multimodal Feature Extraction:** For textual feature extraction, we utilize a CNN to obtain the textual representations of source text and comments. Given a text $t_i$ in a post $p_i$, we first obtain its representation as $O_{1:L_t}^i = \{o_1^i, o_2^i, ..., o_{L_t}^i\}$ where $o_j^t$ indicates the word embeddings of the $j$-th word in a text $t_i$, and $o \in R^d$ where $d$ donates the dimension size of the word embedding. Then, the word embedding matrix $O$ is fed into a CNN framework to obtain the feature map $S^i = \{S_{i1}, S_{i2}, ..., S_{i(L_t-k+1)}\}$ where $k$ is the size of receptive field. Next, we perform max-pooling on $S^i$ to obtain the $\hat{S}^i = max(S^i)$ and extract the final representation for the source text $t_i$ as $R_t^i = concat(\hat{S}_{k=3}^i, \hat{S}_{k=4}^i, \hat{S}_{k=5}^i)$ through concatenating of different receptive fields. Similarly, we extract the representation for comment $C_i$ as $R_c^i = concat(\hat{S}_{k=3}^i, \hat{S}_{k=4}^i, \hat{S}_{k=5}^i)$.

We utilize ResNet-50 (He et al., 2016) to extract the representation for image $v_i$. Specifically, we first extract the output of the second-to-last layer of ResNet-50 and represent it as $V_r^i$, feeding it into a fully connected layer to obtain the final visual features as $R_v^i = \sigma(W_v * V_r^i)$ with the same dimension size as the textual features.

**Multimodal Feature Fusion:** To capture the interaction among different modalities and enhance cross-modal features, we employ a co-attention (Lu et al., 2019) mechanism. For both textual and visual modalities, we first adopt multi-head self-attention (Vaswani et al., 2017) to enhance the inner feature representation. For instance, given a textual feature $R_t^i$, we adopt $Q_t^i = R_t^i W_t^Q$, $K_t^i = R_t^i W_t^K$, and $V_t^i = R_t^i W_t^V$ to calculate matrix $Q$, $K$, and $V$, respectively, where $W_t^Q$, $W_t^K$, and $W_t^V \in R^{d*\frac{d}{H}}$ are linear transformation where $H$ donates the total number of heads. We obtain the multi-head self-attention feature of textual modality as equation 1 as follows.

$$Z_t^i = (||_{h=1}^H softmax(\frac{Q_t^i K_t^{iT}}{\sqrt{d}})V_t^i)W_t^O \quad (1)$$

where "$||$" donates concatation operation. $h$ refers to the $h$-th head, and $W_t^O \in R^{(d*d)}$ indicates the output of linear transformation.

Similarly, we obtain the multi-head self-attention feature of visual modality as equation 2 as follows.

$$Z_v^i = (||_{h=1}^H softmax(\frac{Q_v^i K_v^{iT}}{\sqrt{d}})V_v^i)W_v^O \quad (2)$$

To extract the co-attention between the textual and visual modalities, we perform a similar self-attention process, replacing $R_t^i$ to $Z_v^i$, $R_t^i$ to $Z_t^i$ to generate $Q_v^i$, $K_t^i$ and $V_t^i$, respectively, and finally obtain the enhanced textual features $Z_{vt}^i$ with visual features as equation 3.

$$Z_{vt}^i = (||_{h=1}^H softmax(\frac{Q_v^i K_t^{iT}}{\sqrt{d}})V_t^i)W_{vt}^O \quad (3)$$

Next, we perform the second co-attention between $Z_{vt}^i$ and $Z_v^i$ to obtain the cross-modality feature $\tilde{Z}_{vt}^i$ and $Z_{tv}^i$ as equation 4 and 5, respectively.

$$\tilde{Z}_{vt}^i = (||_{h=1}^H softmax(\frac{Q_v^i K_{vt}^{i\,T}}{\sqrt{d}})V_{vt}^i)W_{vt}^O \quad (4)$$

$$Z_{tv}^i = (||_{h=1}^H softmax(\frac{Q_t^i K_v^{iT}}{\sqrt{d}})V_v^i)W_{tv}^O \quad (5)$$

Finally, we conduct concatenation for the two enhanced features and the initial comment features to obtain the final multimodal features as $Z^i = concat(\tilde{Z}_{vt}^i, Z_{tv}^i, R_c^i)$.

**Output Layer:** Given that supervised contrast learning (SCL) (Khosla et al., 2017) effectively pulls together representations of the same class while excluding representations from different classes, we incorporate the supervised contrast learning function into our total loss for the rumor detection as equation 6. To enhance the robustness of the proposed model, we introduce projection gradient descent (PGD) (Madry et al., 2017) into textual embeddings when calculating the gradient of textual features in each training iteration and adopt it to calculate adversarial perturbations. Then, we recalculate the gradient on the updated textual features, repeat this process with $m$ times,

and adopt spherical space to limit the range of disturbance. Finally, we accumulate the above adversarial gradient to the original gradient and adopt it for parameter adjustment.

$$L_{Teacher} = (1 - \lambda)L_{CE} + \lambda L_{SCL} \qquad (6)$$

where $\lambda \in \{0, 1\}$ is a hyper parameter, $CE$ donates cross entropy (equation 7), and $SCL$ indicates supervised contrast learning (equation 8).

$$L_{CE} = -\frac{1}{N}\sum_{i=1}^{N}\sum_{c=1}^{C} y_{i,c} log \hat{y}_{i,c} \qquad (7)$$

where $C$ indicates the label type (i.e., rumor and non-rumor), $y_{i,c}$ donates the true label with class type $c$, and $\hat{y}_{i,c}$ refers to the predicted probability with class type $c$.

$$L_{SCL} = \frac{1}{N}\sum_{i \in I}\frac{1}{|P_i|}\sum_{p \in P_i} -log\frac{exp(\frac{Z_i.Z_p}{\Gamma})}{\sum_{a \in A_i} exp(\frac{Z_i.Z_a}{\Gamma})} \qquad (8)$$

where $I$ donates the set of indexes of training samples, $P_i$ embodies the set of indexes of positive samples, $A_i$ refers to the indexes of contracted samples, $Z_i$ stands for the normalization, and $\Gamma$ indicates a temperature parameter to control different categories.

### 3.2.2 Student Model

As mentioned in the Introduction section, existing rumor detection models ignore the incomplete modality problem. It is common for the image to fail to load in a given multimodal post. Due to the lack of visual information, only textual information can be used for rumor detection. To address this issue, we adopt knowledge distillation (Hinton et al., 2015) to perform incomplete rumor detection based on our pre-trained teacher model. The motivation behind knowledge distillation is to leverage the soft labels predicted by the teacher network to guide the learning of the student network and improve its performance. By minimizing the distance between the soft probability distributions of the student and teacher models, as measured by the KL loss (equation 9), we aim to align the predictions of the student model with those of the teacher model. In essence, knowledge distillation serves the purpose of incorporating soft targets associated with the teacher network, which exhibits complex yet superior prediction performance, into the overall Loss function.

This facilitates the training of the student network, which is simplified, possesses lower complexity, and is more suitable for deployment in inference scenarios. The ultimate goal is to achieve effective knowledge transfer.

$$L_{KD}(q^t, q^s, \tau) = \sum_{i=1}^{N} 2\tau^2 KL(\sigma(\frac{q_i^t}{\tau}), \sigma(\frac{q_i^s}{\tau})) \qquad (9)$$

where $q^t$ and $q^s$ donate the output of teacher and student network, respectively, $\sigma$ indicates softmax, $\tau$ refers to scale the temperature of the smoothness of two distributions. A lower value of $\tau$ will sharpen the distribution, leading to an expanded difference between the two distributions. It concentrates the distillation on the maximum output predicted by the teacher network. On the other hand, a higher value of $\tau$ will flatten the distribution, narrowing the gap between the teacher and student networks. This broader distribution concentrates the distillation on the entire output range. Then, the total loss of student model is shown in equation 10.

$$L_{Student} = \alpha L_{KD} + (1 - \alpha)L_{CE} \qquad (10)$$

where CE donates cross entropy, and $\alpha$ is a hyper parameter.

## 4 Experiments

### 4.1 Datasets and Evaluation Metrics

We utilize four benchmark multimodal corpora: the two Chinese datasets (e.g., Weibo-19 (Song et al., 2019), Weibo-17 (Jin et al., 2017)) and the two English corpus (e.g., Twitter (Boididou et al., 2018), Pheme (Zubiaga et al., 2017)). Each dataset comprises source text and images. The Weibo-19 and Pheme have comments, while the Weibo-17 and Twitter don't have comments. Table 1 presents the statistics of the four benchmark corpora. We adopt four popular evaluation metrics: i.e., accuracy, precision, recall, and F1-Score, to investigate the performance of our proposed framework and other comparing approaches.

**Baselines**: We take seven baseline models as illustrated in Appendix A.1.

**Hyper-parameter Settings**: Following the approach of existing baseline systems, we divide the dataset into training, validation, and testing sets using a ratio of 7:1:2, respectively. For word

Table 1: Corpus statistics. N: Non-rumors; R: Rumors.

|  | #N | #R | #Images | #Comments |
|---|---|---|---|---|
| Weibo-19 | 877 | 590 | 1,467 | 4,534 |
| Weibo-17 | 4,749 | 4,779 | 9,528 | 0 |
| Twitter | 6,026 | 7,898 | 514 | 0 |
| Pheme | 1,428 | 590 | 2,018 | 21,564 |

embeddings, we employ Word2Vec-style embeddings as proposed in (Yuan et al., 2019). The number of head $H$ in self-attention is set to 8. We adopt Adam (Kingma and Ba, 2014) to optimize our loss function. The learning rate is set to 0.002, and the batch size is set to 64. The value of dropout is set to 0.6. The $\tau$ in knowledge distillation is set to 5.0. The $\Gamma$ in supervised contrastives learning is set to 0.5. The length of source text $L_t$ and comment $L_c$ are set to 50. $\alpha$ is set to 0.7, and $\lambda$ is set to 0.5. The number of $m$ in resistance disturbance is set to 3. We perform 5 runs throughout all experiments and report the average results and standard deviation results.

### 4.2 Results and Discussion

**Model Comparison:** Tables 2 and 3 present the average performance and standard deviation obtained from five executions on both the Chinese and English datasets. On the Pheme and Weibo-19 datasets, since we used the same training, validation, and testing splits as the baseline systems, we directly compare our results with theirs. In addition, we execute the public available source codes from these baselines on the Twitter and Weibo-17 datasets. Since EBGCN and GLAN constructe a propagation graph by using comments in the source posts, we don't report their performance on the Weibo-17 and Twitter datasets without comments information. It is evident from Tables 2 and 3 that our CLKD-IMRD outperforms other models in terms of accuracy, precision, recall, and F1-Score measures. This highlights the significance of multimodal feature fusion and contrastive learning in our approach. While ChatGPT has proven effective in various NLP tasks, its performance in rumor detection is not satisfactory. On the Weibo-17 and Twitter datasets, only short source texts are utilized without comments as a supplementation, resulting in poor performance of the ChatGPT on the rumor detection task. Based on the observations from Tables 2 and 3, we can derive the following insights:

(1) Among the three multi-modal baselines (EANN, MVAE, and SAFE), SAFE achieves the highest performance in all four measures: accuracy, precision, recall, and F1-Score. On the other hand, MVAE demonstrates the poorest performance across all four measures on the Weibo dataset, which highlights the ineffectiveness of the superficial combination of textual and visual modalities in MVAE. In contrast, the incorporation of event information in the EANN model proves beneficial for debunking rumors. Notably, the SAFE model successfully incorporates a deep interaction between textual and visual modalities, resulting in superior performance.

(2) Among the three social graph-based baselines (EBGCN, GLAN, and MFAN), they demonstrate better performance compared to the simpler EANN and MVAE models. Both EBGCN and GLAN achieve comparable performance as they incorporate structural information. However, MFAN, which combines textual, visual, and social graph-based information, outperforms the others in all four measures: accuracy, precision, recall, and F1-Score.

**Performance of Knowledge Distillation:** CLKD-IMRD involves adopting a multimodal contrastive learning model as the teacher model, and multimodal and incomplete modal models as the student models. Such teacher-student framework allows us to transfer knowledge from the multimodal teacher model to both multimodal and single-modal student models. We explore four student models, each utilizing only cross-entropy loss as the loss function.

- **Student-1**: The model incorporates all modalities, including source text, visual information, and user comments.

- **Student-2**: The model focuses on textual (source text) and visual modalities.

- **Student-3**: The model relies on the textual modality, considering both the source text and comments.

- **Student-4**: The model exclusively relies on the source text modality.

Limited to space, the knowledge distillation results on the Weibo-19 and Pheme datasets are shown in Table 4, which indicate that all student models exhibit improvement when guided by the teacher model. Even student-4, which only includes the source text modality, demonstrates a 1.0%-1.7% enhancement in accuracy

Table 2: Performance comparison of rumor detection models on the two Chinese datasets.

| | Weibo-19 | | | | Weibo-17 | | | |
|---|---|---|---|---|---|---|---|---|
| | Accuracy | Precision | Recall | F1-Score | Accuracy | Precision | Recall | F1-Score |
| EANN | 80.96±2.26 | 80.19±2.37 | 79.86±2.46 | 79.87±2.40 | 82.83±2.64 | 82.76±2.42 | 82.68±2.58 | 82.71±2.43 |
| MVAE | 71.67±0.89 | 70.52±0.95 | 70.21±1.01 | 70.34±0.98 | 81.23±1.25 | 84.54±1.75 | 74.68±1.32 | 79.94±1.58 |
| SAFE | 84.95±0.85 | 84.98±0.82 | 84.95±0.91 | 84.96±0.86 | 75.56±1.43 | 76.54±1.35 | 74.95±1.51 | 74.80±1.46 |
| EBGCN | 83.14±2.01 | 85.46±2.12 | 81.76±1.54 | 81.45±1.74 | - | - | - | - |
| GLAN | 82.44±2.02 | 82.45±2.26 | 80.86±1.71 | 81.26±1.93 | - | - | - | - |
| MFAN | 88.95±1.43 | 88.91±1.60 | 88.13±1.68 | 88.33±1.53 | 88.23±1.83 | 88.56±1.91 | 88.07±1.98 | 88.17±1.73 |
| ChatGPT | 29.83±0 | 28.27±0 | 28.95±0 | 28.52±0 | 0±0 | 0±0 | 0±0 | 0±0 |
| CLKD-IMRD | **93.36±1.12** | **93.38±1.23** | **92.91±1.28** | **93.07±1.20** | **91.25±1.01** | **91.29±1.32** | **91.34±1.33** | **91.14±1.71** |

Table 3: Performance comparison of rumor detection models on the two English datasets.

| | Twitter | | | | Pheme | | | |
|---|---|---|---|---|---|---|---|---|
| | Accuracy | Precision | Recall | F1-Score | Accuracy | Precision | Recall | F1-Score |
| EANN | 71.52±1.96 | 77.95±2.14 | 70.65±3.23 | 72.79±2.69 | 77.13±0.96 | 71.39±1.07 | 70.07±2.19 | 70.44±1.69 |
| MVAE | 79.91±0.96 | 84.88±1.23 | 71.48±1.10 | 71.57±1.25 | 77.62±0.64 | 73.49±0.81 | 72.25±0.90 | 72.77±0.81 |
| SAFE | 76.49±1.24 | 82.18±1.62 | 79.50±1.01 | 79.68±0.98 | 81.49±0.84 | 79.88±1.22 | 79.50±0.81 | 79.68±0.70 |
| EBGCN | - | - | - | - | 82.99±0.65 | 81.31±0.73 | 79.29±0.71 | 79.82±0.64 |
| GLAN | - | - | - | - | 83.32±1.64 | 81.25±2.06 | 77.13±3.26 | 78.51±2.68 |
| MFAN | 86.08±1.12 | 85.03±1.67 | 83.61±1.93 | 84.36±1.64 | 88.73±0.83 | 87.07±1.41 | 85.61±1.65 | 86.16±1.04 |
| ChatGPT | 0±0 | 0±0 | 0±0 | 0±0 | 34.29±0 | 24.26±0 | 26.94±0 | 25.53±0 |
| CLKD-IMRD | **88.01±0.91** | **88.15±1.03** | **84.64±0.65** | **86.53±0.97** | **89.09±0.47** | **87.46±0.96** | **85.86±0.16** | **86.58±0.41** |

and F1-Score measures. Similar improvements are observed in the other three student models (student-1, student-2, and student-3). Among these, student-1, utilizing all modalities (source text, visual information, and user comments), achieves the best performance across all four measures. Generally, student-2 outperforms student-3 due to its incorporation of both textual and visual modalities, while student-3 relies solely on textual information.

**Ablation Study:** Limited to space, Table 5 presents the performance of ablation analysis on the Weibo-19 and Pheme datasets, where we examine the impact of various components by considering five cases:

- **w/o text**: We exclude the use of source text.

- **w/o image**: We omit the utilization of image information.

- **w/o comment**: We disregard the inclusion of comments.

- **w/o contrastive learning**: We eliminate the application of contrastive learning.

- **w/o projection gradient descent**: We do not employ projection gradient descent.

Based on the findings in Table 5, several conclusions can be drawn. 1) The source text plays a crucial role in rumor detection. The performance significantly deteriorates when the source text is excluded, underscoring the importance of the textual

modality in identifying rumors. 2) Both images and comments contribute to debunking rumors, as evidenced by their absence leading to a decline in performance. 3) The integration of supervised contrastive learning enhances the model's ability to distinguish between positive and negative samples in the corpora, which positively impacts the performance of the model. 4) The inclusion of projection gradient descent during the adversarial perturbations training phase improves the robustness of our proposed model.

**Impact of Co-attention Settings:** We further analyze the performance comparison with different number of co-attention as illustrated in Appendix A.2.

### 4.3 Impact of Number of Comments

We further analyze the performance of different comment scenarios, considering the following six cases:

- **0% comment**: No comments are used in this case.

- **only the first comment**: Only the first comment is considered.

- **20% comments**: We include 20% of the comments in a time-sequential manner.

- **50% comments**: We include 50% of the comments in a time-sequential manner.

Table 4: Performance comparison of knowledge distillation on Weibo-19 and Pheme. Performance improvement is represented by numbers in parentheses with ↑ symbol.

| | Weibo-19 | | | | Pheme | | | |
| --- | --- | --- | --- | --- | --- | --- | --- | --- |
| | Accuracy | Precision | Recall | F1-Score | Accuracy | Precision | Recall | F1-Score |
| Teacher | 94.24 | 94.07 | 93.95 | 94.01 | 89.61 | 88.49 | 85.92 | 87.06 |
| Student-1 initial | 93.22 | 93.33 | 92.55 | 92.90 | 88.31 | 86.72 | 84.49 | 85.48 |
| Student-1 after KD | 94.24(↑1.02) | 94.67(↑1.34) | 93.40(↑0.85) | 93.93(↑1.03) | 88.83(↑0.52) | 87.39(↑0.67) | 85.11(↑0.62) | 86.13(↑0.65) |
| Student-2 initial | 91.53 | 90.99 | 91.67 | 91.27 | 86.49 | 84.63 | 81.91 | 83.08 |
| Student-2 after KD | 93.22(↑1.69) | 93.67(↑2.68) | 92.28(↑0.61) | 92.85(↑1.58) | 88.31(↑1.82) | 86.23(↑1.60) | 85.26(↑3.35) | 85.72(↑2.64) |
| Student-3 | 92.20 | 92.33 | 91.42 | 91.82 | 87.01 | 84.56 | 83.86 | 84.18 |
| Student-3 after KD | 93.22(↑1.02) | 93.19(↑0.86) | 92.69(↑1.27) | 91.92(↑0.10) | 88.31(↑1.30) | 86.91(↑2.35) | 84.23(↑0.37) | 85.40(↑1.22) |
| Student-4 | 90.85 | 91.85 | 89.33 | 90.23 | 86.23 | 83.65 | 82.76 | 83.18 |
| Student-4 after KD | 92.20(↑1.35) | 92.33(↑0.48) | 91.42(↑2.09) | 91.81(↑1.58) | 87.79(↑1.56) | 86.65(↑3.00) | 83.08(↑0.32) | 84.57(↑1.39) |

Table 5: Ablation study.

| | Weibo-19 | | | | Pheme | | | |
| --- | --- | --- | --- | --- | --- | --- | --- | --- |
| | Accuracy | Precision | Recall | F1-Score | Accuracy | Precision | Recall | F1-Score |
| CLKD-IMRD | 94.24 | 94.07 | 93.95 | 94.01 | 89.61 | 88.49 | 85.92 | 87.06 |
| w/o text | 81.02 | 80.36 | 80.00 | 80.17 | 77.14 | 72.76 | 68.56 | 69.89 |
| w/o image | 92.20 | 92.49 | 91.29 | 91.79 | 88.05 | 86.30 | 84.30 | 85.20 |
| w/o comment | 92.54 | 93.36 | 91.30 | 92.08 | 86.75 | 84.43 | 83.12 | 83.73 |
| w/o contrastive learning | 93.22 | 93.33 | 92.55 | 92.90 | 88.31 | 86.72 | 84.49 | 85.48 |
| w/o projection gradient descent | 92.54 | 92.17 | 92.39 | 92.27 | 87.27 | 84.60 | 84.78 | 84.79 |

Table 6: Performance comparison with different number of comments.

| | Accuracy | Precision | Recall | F1-Score |
| --- | --- | --- | --- | --- |
| **Weibo-19** | | | | |
| 0% comments | 92.54 | 93.36 | 91.30 | 92.08 |
| only the first comment | 94.24 | 94.07 | 93.95 | 94.01 |
| 20% comments | 93.56 | 93.61 | 92.97 | 93.26 |
| 50 % comments | 92.88 | 92.37 | 93.08 | 92.67 |
| 80 % comments | 92.88 | 92.90 | 92.27 | 92.55 |
| all comments | 93.22 | 93.33 | 92.55 | 92.90 |
| **Pheme** | | | | |
| 0% comments | 86.75 | 84.43 | 83.12 | 83.73 |
| only the first comment | 89.61 | 88.49 | 85.92 | 87.06 |
| 20% comments | 87.27 | 84.71 | 82.53 | 84.62 |
| 50 % comments | 87.01 | 85.14 | 82.79 | 83.82 |
| 80 % comments | 88.02 | 87.04 | 83.14 | 84.74 |
| all comments | 88.02 | 85.50 | 85.50 | 85.50 |

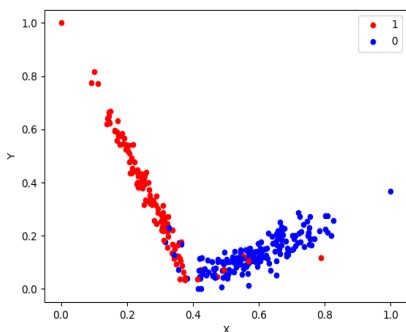

Figure 2: T-SNE visualization on the Weibo-19.

- **80% comments**: We include 80% of the comments in a time-sequential manner.

- **all comments**: All comments are included.

Limited to space, the impact of number of comments results on the Weibo-19 and Pheme are shown in Table 6. Based on the findings in Table 6, we can draw the conclusion that increasing the number of comments does not contribute significantly to debunking rumors. In fact, as the number of comments increases, the introduction of noise becomes more prominent. Interestingly, the first comment proves to be more valuable in the context of rumor detection, as it carries more relevant information for distinguishing between rumors and non-rumors.

**Visualization Studies:** Figures 2 and 3 display the T-SNE visualization of the test data from Weibo-19 and Pheme, respectively. The visualizations clearly depict the successful classification of most samples into distinct groups, demonstrating the effectiveness and strong representation capability of our proposed model.

Figures 4 and 5 showcase the attention visualization samples with the label "non-rumor" and "rumor" from the Weibo-19 and Pheme, respectively, which provides insights into the interaction between textual and visual information, and how the enhanced features contribute to debunking rumors. In Figure 4, the words "cat" and "dog" highlighted in red demonstrate high attention weights and align well with specific regions in the corresponding image. This accurate alignment con-

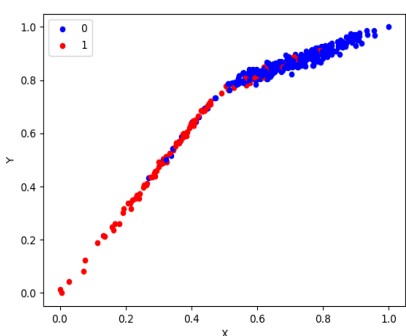

Figure 3: T-SNE visualization on the Pheme.

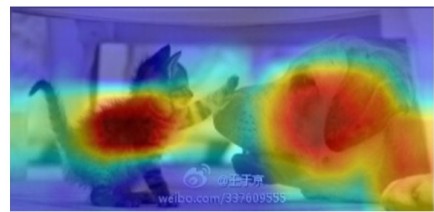

残疾猫在玩伴狗患癌死亡后绝食去世。

Disabled cat die from hunger strike after its playmate dog dies from cancer.

**Label**: Non-rumor.

Figure 4: Attention visualization on the Weibo-19.

tributes to the prediction of the sample as a non-rumor. In contrast, in Figure 5, the words "suspect" and "MartinPlace" fail to align with their respective image regions, indicating poor alignment and predicting the sample as a rumor correctly. These observations highlight the deep semantic interaction between the textual and visual modalities within our proposed model.

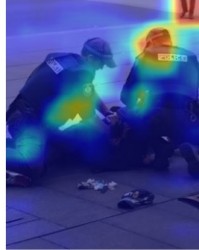

Police take down suspect in MartinPlace. Unclear whether related to siege.

**Label**: rumor.

Figure 5: Attention visualization on the Pheme.

# 5 Conclusion

In this paper, we propose a rumor detection framework that combines supervised contrastive learning and knowledge distillation. Our framework leverages hierarchical co-attention to enhance the representation of textual (source text and comments) and visual modalities, enabling them to complement each other effectively. The utilization of contrastive learning has proven to be successful in debunking rumors. Additionally, knowledge distillation has demonstrated its efficacy in handling incomplete modalities for rumor detection. Moving forward, our future work aims to integrate graph structures, such as social graphs, into our proposed framework for further improvement.

## Acknowledgments

The authors would like to thank anonymous reviewers for their insightful comments on this paper. This research was supported by the National Natural Science Foundation of China under Grant 62162031, the Natural Science Fund project in Jiangxi province under Grant 20224ACB202010, and the National Natural Science Foundation of China under Grant 62266023.

## Limitations

(1) Limited generalizability: Our experiments were conducted on specific datasets (Chinese Weibo and English Pheme/Twitter) and may not fully represent the characteristics of other rumor detection scenarios or platforms. Generalizability to different datasets and languages needs to be further explored.

(2) Absence of real-time evaluation: Our evaluation primarily focused on offline performance measures, and we did not consider real-time or dynamic evaluation scenarios. Future work should investigate the model's performance in real-time rumor detection settings.

## Ethics Statement

The benchmark datasets utilized in the project primarily reflect the culture of the English-speaking and Chinese-speaking populace. Socio-economic biases may exist in the public and widely-used datasets, and models trained on these datasets may propagate this biases.

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

## A  Appendix

### A.1  Baselines

To investigate the performance of our proposed CLKD-IMRD model, we will perform comparison studies on following approaches.

**EANN** (Wang et al., 2018): A GAN-based multimodal model that adopts characteristics of the invariance of an event to facilitate the detection of the newly arrived events of fake news.

**MVAE** (Khattar et al., 2019): A variational auto-encoder-based model that captures the shared representation between textual and visual modalities.

**SAFE** (Zhou et al., 2020): A similarity-aware multimodal model that debunks fake news from the similarity between multimodal and cross-modal features jointly.

**EBGCN** (Wei et al., 2021): An edge-enhanced bayesian graph convolutional networks-based model that investigates the reliability of potential relationships in propagation structures.

**GLAN** (Yuan et al., 2019): An integration of local semantic and global structural information-based model that debunks rumor.

**MFAN** (Zheng et al., 2022): A feature-enhanced attention networks-based multimodal model that combines textual, visual, and social graphs to enhances graph topology and neighborhood aggregation processes when detecting rumor.

**ChatGPT** [4]: A popular application showcasing the capabilities of the GPT language model is our baseline model. Since ChatGPT cannot receive image modality, we adopt the source text and the first comment as the input of ChatGPT, along with a question "judge it a rumor or not" to obtain the response, and map the results to labels (i.e.g, "yes" to rumor, "no" to non-rumor, "unable to judge" to none).

### A.2  Impact of Co-attention Settings

Limited to space, Table 7 lists the performance comparison with different number of co-attention on the Weibo-19 and Pheme datasets. We consider four cases as follows.

- **Zero Co-attention**: In this case, no co-attention is used. The representations of the source text, visual images, and comments are directly concatenated.

Table 7: Performance comparison with different number of co-attention; co-att donates co-attention.

|  | Accuracy | Precision | Recall | F1-Score |
|---|---|---|---|---|
| **Weibo-19** | | | | |
| Zero co-att | 92.54 | 92.04 | 92.66 | 92.31 |
| One co-att | 93.22 | 93.06 | 92.82 | 92.94 |
| Two co-att | 93.90 | 93.77 | 93.53 | 93.64 |
| Three co-att | 94.24 | 94.07 | 93.95 | 94.01 |
| **Pheme** | | | | |
| Zero co-att | 86.49 | 83.31 | 85.79 | 84.33 |
| One co-att | 88.57 | 87.58 | 84.15 | 85.60 |
| Two co-att | 88.83 | 87.80 | 84.59 | 85.96 |
| Three co-att | 89.61 | 88.49 | 85.92 | 87.06 |

- **One Co-attention**: Here, only the first co-attention is employed.

- **Two Co-attention**: In this case, two text-visual co-attention operations are conducted. The enhanced textual-visual representation is then concatenated with the comment representation.

- **Three Co-attention**: This case involves the adoption of all three co-attention operations. The enhanced textual and visual representations are concatenated with the comment representations.

From Table 7, we observe that the performance improves with an increase in the number of co-attentions. Specifically, the Zero co-attention case demonstrates the lowest performance across all three measures (accuracy, precision, F1-Score), which indicates the importance of capturing the deep interaction between textual and visual modalities through co-attentions. With the addition of one co-attention, we observe an improvement in performance as the enhanced textual representation aids in debunking rumors. As expected, the best performance is achieved when both the enhanced textual and visual representations are utilized, as evidenced by their superior results across all four measures.

---

[4]https://openai.com/blog/chatgpt