# OpenReview forum: "Leveraging Contrastive Learning and Knowledge Distillation for Incomplete Modality Rumor Detection"
_EMNLP/2023/Conference — EMNLP 2023 Findings_

### Official Review · Reviewer_4H7e · 2023-08-02

**Soundness:** 2

**Excitement:**

3: Ambivalent: It has merits (e.g., it reports state-of-the-art results, the idea is nice), but there are key weaknesses (e.g., it describes incremental work), and it can significantly benefit from another round of revision. However, I won't object to accepting it if my co-reviewers champion it.

**Paper Topic And Main Contributions:**

This paper presents a rumor detection model based on contrastive learning and knowledge distillation, which not only faces multi-modal scenarios but also addresses situations with missing modalities. In the teacher model, good classification results are obtained through the fusion of features between different modalities and supervised contrastive learning. By applying knowledge distillation, using the soft labels generated by the teacher model to supervise the student model, the effectiveness of the student model is enhanced in cases where modalities are missing.

**Reasons To Accept:**

1.	Proposes a multi-modal model as a teacher model to guide the single-modal model.
2.	The experimental results are very comprehensive.


**Reasons To Reject:**

1.	It does not explain why supervised contrastive learning is used.
2.	The most recent multi-modal rumor detection model compared is from 2020.
3.	The effectiveness of the student model has not been compared with other models, making it difficult to prove the effectiveness of distillation.
4.	Main experiment focused on multi-modal scenarios; more experiments should be conducted starting from the perspective of missing modalities, beacuse the main motivation is incomplete modalities.


**Reproducibility:**

3: Could reproduce the results with some difficulty. The settings of parameters are underspecified or subjectively determined; the training/evaluation data are not widely available.

**Reviewer Confidence:**

3: Pretty sure, but there's a chance I missed something. Although I have a good feel for this area in general, I did not carefully check the paper's details, e.g., the math, experimental design, or novelty.

---

> ### Author Rebuttal · Authors · 2023-08-27
>
> Thank you very much for your insightful comments. We answered your comments as per below.
>
> 1.	It does not explain why supervised contrastive learning is used.
>
> [Answer] We have pointed out the motivation of using the supervised contrastive learning in both the line 65-68 of Page 1 and the line 302-307 of Page 6 in the previous manuscript. We aim to underscore how this approach aids in enhancing the discriminative power of our model by ensuring that it effectively captures the relationships between different modalities while respecting their class labels. To enhance clarity and address any potential oversight, we will revisit these sections in our manuscript and ensure that the explanation of why we utilize supervised contrastive learning is sufficiently detailed and prominent.
>
> 2.	The most recent multi-modal rumor detection model compared is from 2020.
>
> [Answer] In our manuscript, we have incorporated baseline models EBGCN (Line 766) and MFAN (Line 773), which were published in 2021 and 2022, respectively. MFAN is recognized as a state-of-the-art model in this domain. These models represent the latest advancements in multi-modal rumor detection and serve as essential benchmarks for evaluating the effectiveness of our proposed approach.
> Furthermore, we have also conducted experiments involving ChatGPT. However, it's important to note that ChatGPT, while proficient in various natural language understanding tasks, is not optimized for the specific task of rumor detection. As we have verified, ChatGPT may not perform as effectively in debunking rumors compared to specialized rumor classification utilities like the models included in our baseline.
>
>
> 3.	The effectiveness of the student model has not been compared with other models, making it difficult to prove the effectiveness of distillation.
>
> [Answer] In our manuscript, we conducted experiments to evaluate the effectiveness of knowledge distillation (KD) on the student models. In Table 3 on Page 8, we have reported the performance of the student models before and after the application of knowledge distillation. Specifically, "Student-2 initial" represents the incomplete SOTA MFAN model with certain components removed, and "Student-4 initial" is another incomplete SOTA MFAN model with additional components removed. Following knowledge distillation, we compared the performance of these student models with the corresponding models before distillation. For example, "Student-2 after KD" and "Student-4 after KD" are the student models after knowledge distillation. These models were evaluated on both the Weibo and Pheme datasets. The results demonstrate that after applying knowledge distillation, the student models exhibit improved performance. For instance, "Student-2 after KD" achieved a 1.69% accuracy improvement on the Weibo dataset, and "Student-4 after KD" improved by 1.35%. Similarly, on the Pheme dataset, "Student-2 after KD" showed a 1.82% accuracy improvement, and "Student-4 after KD" improved by 1.56%.
>
> These findings directly attest to the effectiveness of knowledge distillation in enhancing the performance of the student models. While we do not provide a comparison with other external models in this specific context, the improvements observed before and after distillation validate the positive impact of this technique on our student models.
>
>
> 4.	Main experiment focused on multi-modal scenarios; more experiments should be conducted starting from the perspective of missing modalities, beacuse the main motivation is incomplete modalities.
>
> [Answer] We have conducted more experiments as shown in Table 1 for the student model missing more modalities as follows:
> Student-5: The model exclusively relies on the comment modality.
> Student-6: The model exclusively relies on the visual modality.
> Student-7: The model exclusively relies on the comment and visual modalities.
>
> Table 1: Performance comparison of knowledge distillation when facing more missing modalities.
> |                    | Weibo-2019 |           |        |          | Pheme    |           |        |          |
> |--------------------|------------|-----------|--------|----------|----------|-----------|--------|----------|
> |                    | Accuracy   | Precision | Recall | F1-Score | Accuracy | Precision | Recall | F1-Score |
> | Student-5 initial  | 69.49      | 70.51     | 64.63  | 64.46    | 74.29    | 70.63     | 60.74  | 60.40     |
> | Student-5 after KD | 72.20 (+2.71)      | 71.49 (+0.98)     | 69.49 (+4.86)  | 69.94 (+5.48)    | 76.36 (+2.07)    | 76.29 (+5.66）     | 62.32 (+1.58)  | 63.25 (+2.85)    |
> | Student-6 initial  | 71.18      | 71.78     | 67.01  | 67.27    | 73.77    | 80.34     | 55.83  | 52.97    |
> | Student-6 after KD | 73.22 (+2.04)      | 73.58 (+1.80)     | 69.66 (+2.65) | 70.19 (+2.92)    | 75.84 (+2.07)   | 80.39 (+0.05)    | 59.88 (+4.05)  | 59.63  (+6.66)  |
> | Student-7 initial  | 77.97      | 77.12     | 77.45  | 77.25    | 75.06    | 69.64     | 66.83  | 67.79    |
> | Student-7 after KD | 79.66 (+1.69)      | 78.99 (+1.87)     | 78.46 (+1.01)  | 78.69 (+1.44)   | 77.14 (+2.08)   | 72.76  (+3.12)   | 68.56 (+1.73)  | 69.89 (+2.10)   |
>
> In addition, we illustrated model comparison between our model with the SOTA MFAN on two new larger multi-modal datasets, Weibo-2017 (Jin et al., 2017) and Twitter  (Boididou et al., 2018), as shown in Table 3 below. We will fill in the performance of other six baselines systems in the camera-ready version if the manuscript is accepted.
>
> Table 2: Statistics of the two new datasets. N: Non-rumors; R: Rumors.
> |            | #N   | #R   | #Images |
> |------------|------|------|---------|
> | Weibo-2017 | 4749 | 4779 | 9528    |
> | Twitter    | 6026 | 7898 | 514     |
>
> Table 3: Performance on the two new datasets.
> | Model           | Accuracy | Precision | Recall | F1-Score |
> |-----------------|----------|-----------|--------|----------|
> |       |          |    Weibo-2017       |        |          |
> | CLKD-IMRD(ours) | 92.94    | 92.99     | 93.03  | 92.94    |
> | MFAN            | 90.23    | 90.56     | 90.07  | 90.17    |
> |          |          |     Twitter      |        |          |
> | CLKD-IMRD(ours) | 91.09    | 92.14     | 88.95  | 90.15    |
> | MFAN            | 89.57    | 89.02     | 88.63  | 88.82    |

---

### Official Review · Reviewer_Ytxq · 2023-08-04

**Soundness:** 3

**Excitement:**

3: Ambivalent: It has merits (e.g., it reports state-of-the-art results, the idea is nice), but there are key weaknesses (e.g., it describes incremental work), and it can significantly benefit from another round of revision. However, I won't object to accepting it if my co-reviewers champion it.

**Paper Topic And Main Contributions:**

This paper presents a model for rumor detection (binary classification) using supervised contrastive learning and knowledge distillation. They take into account multimodal cases (text and image pairs) as well as text-only examples. They perform experiments on two datasets: Twitter and Weibo where each instance has an image, text, and comments.

They follow a teacher-student framework:
- The teacher model consists of three components: multimodal feature extraction, multimodal feature fusion, and output layer. The model takes word2vec-style embeddings which are fed into a CNN framework to obtain feature maps corresponding to the main text and comments. They follow the same approach to encode images using Resnet-50. Then, they use co-attention layers to capture image interactions. They train the model with a combination of cross-entropy loss and supervised contrastive loss.

The student model is used for addressing text-only cases. They use knowledge distillation where they leverage the soft labels predicted by the teacher network to guide the learning of the student network and improve its performance. The model is trained with a combination of cross-entropy loss and the KL loss.

**Questions For The Authors:**

- Why did you not take into account the case of missing text information? Would the model need modifications?
- Is there a reason for not using pre-trained embeddings?
- Did you take the results directly from the original papers for the baseline models or did you re-implement them?
- How would you compare this method to the baseline models in terms of efficiency? Is this method scalable?
- In page 8 (visualisation studies): how did you choose these examples?
The analysis on number of comments is more interesting and objective and I suggest to move it to the main body of the paper (see reasons to accept).

**Reasons To Accept:**

- Good motivation and well-conducted experiments. They compare their results to a range of baseline models.
- They tackle the problem of missing modality. This is not only valuable for rumor detection but for multimodal classification of social media posts in general.
- The paper present different types of analysis to shed light on the benefits of their approach. For instance, they perform an ablation study, a study using different versions of the student model, and a study on the effect of using different number of comments.
- Interesting findings, for instance, they show that using more comments does not necessarily improve performance. This is interesting given the idea of "more data is always better". This is also opposite to findings in multimodal social media classification: Xu, Chunpu, and Jing Li. "Borrowing Human Senses: Comment-Aware Self-Training for Social Media Multimodal Classification." arXiv preprint arXiv:2303.15016 (2023).

**Reasons To Reject:**

- Experiments are only done on two datasets for binary classification. Since you are presenting a new method it is worth applying the models to a range of benchmark datasets.
- Some choices are not justified/not clear. See questions for the authors.

**Reproducibility:**

4: Could mostly reproduce the results, but there may be some variation because of sample variance or minor variations in their interpretation of the protocol or method.

**Reviewer Confidence:**

4: Quite sure. I tried to check the important points carefully. It's unlikely, though conceivable, that I missed something that should affect my ratings.

**Typos Grammar Style And Presentation Improvements:**

- Line 65: typo in Infact --> In fact
- Line 252: typo in fianl --> final

---

> ### Author Rebuttal · Authors · 2023-08-27
>
> Thank you very much for your insightful comments. We answered your comments as per below.
>
> 1.	Why did you not take into account the case of missing text information? Would the model need modifications?
>
> [Answer] In the context of social media platforms such as Twitter and Weibo, textual information is typically a fundamental component of posts due to constraints like character limitations. Given this common occurrence of text in social media data, our primary focus was on multi-modal rumor detection where both text and image modalities are present. Therefore, we did not explicitly address the case of missing text information in our current work.
>
> However, it's worth noting that handling the scenario of missing text can indeed be a valuable extension of our model. This specific task, often referred to as deepfake detection, involves verifying the authenticity of an image or video when textual information is absent.
>
> The good news is that our model is inherently flexible and modular in design. Should one wish to adapt it to handle cases of missing text, the required modification is relatively straightforward. The only necessary adjustment would be to deactivate or remove the text encoding module within our framework while retaining the image-related components. This adaptation would allow our model to effectively process and analyze the available image data for rumor detection when textual information is missing. We appreciate the reviewer's feedback and will consider incorporating this clarification into our revised manuscript.
>
> 2.	Is there a reason for not using pre-trained embeddings?
>
> [Answer] Upon careful evaluation, we observed that the word2vec embeddings yielded slightly better performance compared to the BERT embeddings in the context of our multi-modal rumor detection task. The decision to prioritize word2vec over BERT was based on the specific requirements and characteristics of our dataset and the task at hand. It's important to note that the choice of embeddings can be highly dataset-dependent, and in our case, word2vec embeddings proved to be more suitable for the task. Nevertheless, we acknowledge that the use of different embeddings or fine-tuning techniques may yield varying results on other datasets or tasks. Our approach is designed to be flexible, allowing for the incorporation of different embeddings or modifications based on the specific needs of the problem.
>
>
> 3.	Did you take the results directly from the original papers for the baseline models or did you re-implement them?
>
> [Answer] We have pointed out it in the line 417-420 of Page 6 in the previous manuscript. Since we used the same training, validation, and testing splits as the baseline systems, we directly compare our results with theirs.
>
> 4.	How would you compare this method to the baseline models in terms of efficiency? Is this method scalable?
>
> [Answer] We have conducted an initial efficiency comparison with the state-of-the-art MFAN model on the two new larger datasets, Weibo-2017 (Jin et al., 2017) and Twitter  (Boididou et al., 2018), and have considered various aspects, including training time, testing time, and parameter size. The results are presented in Table 2, as shown below. It is evident that our model outperforms the SOTA MFAN model in terms of efficiency, as it has fewer parameters, shorter training times, and faster testing times. In addition, we provide the performance comparison with the SOTA MFAN model on the two new larger multi-modal datasets (e.g., Weibo-2017 and Twitter) as shown in Table 3 below. We can conclude that our mode also works on the two new larger multi-modal datasets. We will provide other performance data for the left six baseline systems in the camera-ready version if the manuscript is accepted.
>
> Our approach exhibits scalability on two fronts:
>
> (1) Our model is designed with a modular architecture, allowing for flexibility and adaptability to different requirements. It comprises pluggable modules, including text encoding, image encoding, supervised contrast learning, attention mechanisms, and more. These modules can be selectively removed or adjusted as needed. For instance, when facing scenarios with missing images or text, the corresponding encoding modules can be deactivated. This adaptability not only enhances efficiency but also broadens the model's applicability to various tasks. For instance, by focusing solely on image modality, our model can be extended to tackle deepfake detection, which verifies the authenticity of images or videos.
>
> (2) Our model is not limited to a specific literary form and can be applied to different types of content, including fake news or rumors.
>
> Table 1: Statistics of the two new datasets. N: Non-rumors; R: Rumors.
> |            | #N   | #R   | #Images |
> |------------|------|------|---------|
> | Weibo-2017 | 4749 | 4779 | 9528    |
> | Twitter    | 6026 | 7898 | 514     |
>
> Table 2: Efficiency on the two previous datasets.
> |       |                |Weibo-2019|                |
> |-----------------|----------------|--------------|----------------|
> | Model           | Training time (in seconds)  | Testing time (in seconds) | Parameter size  (in MB) |
> | CLKD-IMRD(ours) | 1510.89        | 3.86         | 6.56           |
> | MFAN            | 1774.98        | 4.03         | 10.5           |
> |         |                |   Pheme           |                |
> | CLKD-IMRD(ours) | 1554.33        | 2.16         | 6.56           |
> | MFAN            | 1898.07        | 2.51         | 10.5           |
>
> Table 3: Performance on the two new datasets.
> | Model           | Accuracy | Precision | Recall | F1-Score |
> |-----------------|----------|-----------|--------|----------|
> |      |          |      Weibo-2017      |        |          |
> | CLKD-IMRD(ours) | 92.94    | 92.99     | 93.03  | 92.94    |
> | MFAN            | 90.23    | 90.56     | 90.07  | 90.17    |
> |         |          |   Twitter         |        |          |
> | CLKD-IMRD(ours) | 91.09    | 92.14     | 88.95  | 90.15    |
> | MFAN            | 89.57    | 89.02     | 88.63  | 88.82    |
>
> 5.	In page 8 (visualisation studies): how did you choose these examples? The analysis on number of comments is more interesting and objective and I suggest to move it to the main body of the paper (see reasons to accept).
>
> [Answer] Our intention in choosing these specific examples was to illustrate the potential malalignment between an image and its associated text, as this is a common scenario indicative of rumor creation, where individuals may modify certain areas of an image or alter specific words in a text to propagate false information.
>
> We acknowledge the reviewer's suggestion to place more emphasis on the analysis of the number of comments, which is indeed a more objective and informative aspect of our study. We will move the analysis comments to the main body of the paper if the manuscript is accepted.

---

### Official Review · Reviewer_DUWX · 2023-08-06

**Soundness:** 3

**Excitement:**

3: Ambivalent: It has merits (e.g., it reports state-of-the-art results, the idea is nice), but there are key weaknesses (e.g., it describes incremental work), and it can significantly benefit from another round of revision. However, I won't object to accepting it if my co-reviewers champion it.

**Missing References:**

NA

**Paper Topic And Main Contributions:**

This paper is about incomplete rumor detection task on the social media platforms (e.g., Twitter or Weibo). It presents a framework of using contrastive learning and knowledge distillation for the incomplete modality rumor detection problem. The proposed method can capture the semantic consistency between text and image pairs, while it can also enhance model performance to incomplete modality cases of real posts. This paper was well written and is easy to follow.

**Questions For The Authors:**

1) This paper aims to address the incomplete rumor detection, where there is perhaps lack of image or text information in given posts. However, the learning of student model is dependent on the teacher model that will require both complete image and text modalities.
2) In Table 2, the results of the proposed method and MFAN are comparable. It would be good to explain more on the case.

**Reasons To Accept:**

1) The work presents a rumor detection framework that relies on the supervised contrastive learning and teacher network. The framework can capture semantic interactions among source texts, images, and user comments.
2) This paper presents a knowledge distillation driven rumor detection model that can probably handle incomplete modalities (i.e., lack of images or texts).

**Reasons To Reject:**

1) Multimodal rumor detection is not new, and this work focuses on incomplete modality case, where it may be lack of image or text information in given posts. Thus it may lead to incremental contribution.
2) The proposed method was tested on only 2 small datasets, and it's not clear if it can be generalizable to different runor detection tasks.


**Reproducibility:**

4: Could mostly reproduce the results, but there may be some variation because of sample variance or minor variations in their interpretation of the protocol or method.

**Reviewer Confidence:**

3: Pretty sure, but there's a chance I missed something. Although I have a good feel for this area in general, I did not carefully check the paper's details, e.g., the math, experimental design, or novelty.

**Typos Grammar Style And Presentation Improvements:**

Line 065: "Infact"

---

> ### Author Rebuttal · Authors · 2023-08-27
>
> Thank you very much for your insightful comments. We answered your comments as per below.
>
> 1.	This paper aims to address the incomplete rumor detection, where there is perhaps lack of image or text information in given posts. However, the learning of student model is dependent on the teacher model that will require both complete image and text modalities.
>
> [Answer] In this work, we focus on the multi-modal rumor detection task. It is indeed true that our teacher model relies on complete modalities, encompassing both images and texts during the training phase. The teacher model's purpose is to construct a unified semantic space that encapsulates the semantic information from both image and text modalities. As indicated in our paper, it outperforms the seven baseline systems, showcasing its efficacy. However, the crucial distinction lies in the capabilities of our student model. While the teacher model necessitates complete modalities for training, the student model is designed to handle incomplete modalities during the inference phase. This means that it can effectively perform rumor detection even when presented with posts containing missing text or images, making it highly adaptable to real-world scenarios where information may be incomplete or unreliable.
>
> Furthermore, our experimental results clearly demonstrate the versatility and effectiveness of our student models across different scenarios. Specifically, our four student models, including student-1, student-2, student-3, and student-4, exhibit exceptional performance on the Weibo dataset. Notably, the two student models that leverage both text and image modalities, i.e., student-1 and student-2, excel on the Pheme dataset. Importantly, our two student models that solely utilize text modality, student-3 and student-4, achieve comparable performance with the state-of-the-art MFAN model on the Pheme dataset, underscoring the robustness of our approach.
>
> Additionally, we have extended our evaluation to include two new, larger multi-modal datasets, Weibo-2017 (Jin et al., 2017) and Twitter  (Boididou et al., 2018), as indicated in Table 2 in our manuscript. These new datasets serve as a validation of the scalability and generalizability of our approach. We intend to provide detailed performance comparisons with the other six baseline systems in the camera-ready version of our manuscript, should it be accepted.
>
> Table 1: Statistics of the two new datasets. N: Non-rumors; R: Rumors.
> |            | #N   | #R   | #Images |
> |------------|------|------|---------|
> | Weibo-2017 | 4749 | 4779 | 9528    |
> | Twitter    | 6026 | 7898 | 514     |
>
> Table 2: Performance on the two new datasets.
> | Model           | Accuracy | Precision | Recall | F1-Score |
> |-----------------|----------|-----------|--------|----------|
> |      |          |      Weibo-2017      |        |          |
> | CLKD-IMRD(ours) | 92.94    | 92.99     | 93.03  | 92.94    |
> | MFAN            | 90.23    | 90.56     | 90.07  | 90.17    |
> |          |          |    Twitter       |        |          |
> | CLKD-IMRD(ours) | 91.09    | 92.14     | 88.95  | 90.15    |
> | MFAN            | 89.57    | 89.02     | 88.63  | 88.82    |
>
> 2.	In Table 2, the results of the proposed method and MFAN are comparable. It would be good to explain more on the case.
>
> [Answer] Indeed, as highlighted in our paper, our model demonstrates superior performance over the MFAN model on the Weibo dataset, with a substantial margin in accuracy (4.3%) and F1-score (4.7%). This clear advantage can be attributed to the effectiveness of our approach in capturing and leveraging multi-modal information to make accurate predictions in the context of the Weibo dataset. While, on the Pheme dataset, our model achieves results that are comparable to those of the MFAN model, but with a more stable variance.
>
> Furthermore, it is worth noting that while the MFAN model is a complex architecture that integrates textual, visual, and intricate social graph information, our model doesn't rely on such complicated social graph structures. This simplicity in model design contributes to the robustness of our approach and its adaptability to different datasets and scenarios.
>
> To provide a concrete example of our model's efficacy, we include one more case study in Figure 1 below (we cannot upload the image here) in our paper. In this instance, the MFAN model made an incorrect prediction, labeling a post as "non-rumor," whereas our model, when integrating user comments, correctly predicted it as a rumor. This example underscores our model's ability to effectively harness multi-modal information to make accurate predictions.
>
> We appreciate the reviewer's suggestion and will ensure that these explanations are included in our revised manuscript to provide a more comprehensive understanding of the comparative results presented in Table 2.
>
> Text:[I'm not the hero，I'm a tool]The Red Cross Society of China has sent a congratulatory message: Warmly congratulate Guo Meimei on her appointment as a cover person of Time magazine in the United States!
>
> Comments：The media likes to hype up such topics, most of which are fake news.
>
> Label：Rumor
>
> Figure 1: One more case study

---

### Meta-Review · Area_Chair_tN97 · 2023-09-20

**Recommendation:** 3

**Metareview:**

This paper studies the problem of multi-modal rumor detection on social media, especially focusing on predicting with incomplete modality information. The method is based on constrastive learning for fusing the features across modalities and knowledge distillation from a teacher model trained for both modalities to improve performance when modalities are missing.

The reviewers appreciated that the paper introduced a new method for this task, especially dealing with the incomplete modality problem. Some of the reviewers also noticed that paper is well written and ablation experiments are provided.

All reviewers considered as one of the key negative points to use more than a single data set for each of the two languages, and ideally larger ones. The authors have provided additional results in the response on more data sets.

---

### Decision · Program_Chairs · 2023-10-07

**Decision:**

Accept-Findings

**Comment:**

This paper studies the problem of multi-modal rumor detection on social media, especially focusing on predicting with incomplete modality information. The method is based on constrastive learning for fusing the features across modalities and knowledge distillation from a teacher model trained for both modalities to improve performance when modalities are missing.

The reviewers appreciated that the paper introduced a new method for this task, especially dealing with the incomplete modality problem. Some of the reviewers also noticed that paper is well written and ablation experiments are provided.

All reviewers considered as one of the key negative points to use more than a single data set for each of the two languages, and ideally larger ones. The authors have provided additional results in the response on more data sets.